# Automated Detection of Interpretable Causal Inference Opportunities: Regression Discontinuity Subgroup Discovery

Tony Liu [1 2]   Patrick Lawlor [3]   Lyle Ungar [1]   Konrad Kording [1 4]   Rahul Ladhania [5]

## Abstract

Treatment decisions based on cutoffs of continuous variables, such as the blood sugar threshold for diabetes diagnosis, provide valuable opportunities for causal inference. Regression discontinuities (RDs) are used to analyze such scenarios, where units just above and below the threshold differ only in their treatment assignment status, thus providing as-if randomization. In practice however, implementing RD studies can be difficult as identifying treatment thresholds require considerable domain expertise – furthermore, the thresholds may differ across population subgroups (e.g., the blood sugar threshold for diabetes may differ across demographics), and ignoring these differences can lower statistical power. Here, we introduce Regression Discontinuity SubGroup Discovery (RDSGD), a machine learning method that identifies more powerful and interpretable subgroups for RD thresholds. Using a claims dataset with over 60 million patients, we apply our method to multiple clinical contexts and identify subgroups with increased compliance to treatment assignment thresholds. As subgroup-specific treatment thresholds are relevant to many diseases, RDSGD can be a powerful tool for discovering new avenues for causal estimation across a range of clinical applications.

## 1. Introduction

Many questions in data science are ultimately causal in nature, yet evaluating causal questions through experimental randomization can be costly or otherwise infeasible (Musci & Stuart, 2019). There are numerous methods that esti-mate causality from observational data, but many rely on the key assumption of *no unobserved confounding*, which is generally difficult to justify in high-dimensional data settings (Hernán & Robins, 2020). However, econometricians over the past few decades have been developing study designs that can make credible causal claims from observational data (Leamer, 1983). These study designs address confounding by exploiting naturally occurring randomness in the data, so-called *quasi-experiments* (Angrist & Pischke, 2008; Liu et al., 2021).

We focus on the regression discontinuity (RD), a specific quasi-experimental method for evaluating causal effects from observational data where a cutoff in an observed continuous *running variable* determines treatment assignment (Hahn et al., 2001). Such a situation may arise when treatment depends on a threshold. For example, when a patient's blood sugar level (measured by A1C %) is above 6.5%, they are diagnosed as diabetic (American Diabetes Association, 2010) and hence eligible for treatment assignment. Critically, RDs are more robust to confounding than other observational causal inference methods (Lee & Lemieux, 2009), as the cutoff in treatment assignment provides "as-if" randomization for individuals just above and just below the cutoff: a patient with an A1C of 6.5%, on average, is not materially different from a patient with an A1C of 6.4%, yet the former is diagnosed with diabetes and treated for the disease while the latter is not. Because of this "as-if" randomization, RDs allow us to estimate treatment effects at the threshold without explicit randomization. RD opportunities are particularly natural in medicine, where thresholds govern diagnoses or treatment for many diseases, e.g. diabetes, coronary artery disease, and cancer (Petersen et al., 2020; Scott et al., 2022; Oeffinger et al., 2015).

Despite the ubiquity of such RD opportunities in clinical settings, RDs are often underutilized (Moscoe et al., 2015; Marinescu et al., 2018). Because *a priori* knowledge of the treatment threshold is needed to use an RD, the typical study design approach is a "top-down" process, in which a domain expert hypothesizes that a particular data generating process might yield an RD, followed by verification of study validity by examining the data. Often enough, the RD opportunity is underpowered due to sample size limitations (Naidech

[1] Department of Computer and Information Science, University of Pennsylvania. [2] Roblox. [3] Children's Hospital of Philadelphia. [4] Department of Neuroscience, University of Pennsylvania. [5] School of Public Health, University of Michigan. Correspondence to: Tony Liu <liutony@seas.upenn.edu>.

*Workshop on Interpretable ML in Healthcare at International Conference on Machine Learning (ICML)*, Honolulu, Hawaii, USA. 2023. Copyright 2023 by the author(s).

et al., 2020; McKenzie, 2016). Identifying potential RDs is an ad-hoc process that relies heavily on human intuition and domain expertise, and thus does not scale well to the vast amounts of high-dimensional data we have available today.

This holds especially true as treatment thresholds in practice are often multi-faceted, with heterogeneity in *treatment assignment* as a function of other covariates. For example, in medicine, though diagnostic criteria for diabetes ostensibly are made only according to blood sugar levels, the risk for the disease varies by gender, race, and age categories, leading to different clinical decisions where official guidelines may not always be followed. As treatment assignment thresholds become more complex, it becomes difficult for domain experts to generate study designs and verify them. Thus, taking a "bottom-up" data-driven approach would streamline and scale RD study discovery, unlocking more opportunities for causal inference.

Futhermore, these threshold decisions could identify demographic biases in treatment assignment, as opposed to differences in treatment assignment that are medically justified. Previous work has shown that there is implicit bias present in clinical thresholds (FitzGerald & Hurst, 2017), and bottom-up data-driven approaches for detecting candidate RDs can help identify differences in treatment assignment. Thus, data-driven methods for RD discovery have the potential to not only facilitate study design, but also can inform policy in making treatment decisions more equitable.

Here we propose a data-driven method, Regression Discontinuity Subgroup Discovery (RDSGD), to learn RD subgroups with different treatment assignment thresholds (see Figure 1 for an illustration of our approach). We frame regression discontinuity discovery similarly to the task of conditional average treatment effect (CATE) estimation (Section 3). Note that our method differs from CATE estimation by focusing on heterogeneity in treatment *assignment* rather than heterogeneity in treatment *effects*. We introduce a novel statistical framework targeting higher *effective sample sizes* of the discovered subgroups to maximize statistical power and maintain interpretability(Section 4). We show the utility of our approach through both synthetic experiments (Section 5) and a case study using a medical claims dataset consisting of over 60 million patients (Section 6). We apply our method to three clinical contexts (breast cancer screening, colon cancer screening, diabetes diagnosis) in this data and discover subgroups, with some that are validated by clinical domain knowledge and others that show promise as potential studies. RDSGD can not only discover new opportunities for quasi-experimental studies in healthcare but also can provide actionable interpretability in the form of treatment assignment subgroups, which can be easily understood and validated by clinical practitioners (Section 7).

## 2. Related Work

Automatic regression discontinuity discovery has been explored in a number of different contexts. Porter & Yu (2015) propose a statistical testing framework for regression discontinuity treatment effect inference where the discontinuity point is unknown. In particularly relevant work, Herlands et al. (2018) define an automated RD search procedure called local regression discontinuity discovery (LoRD3) that first requires fitting a "smooth" background function to the probability of treatment, and then computing test statistics for candidate RDs based on the residuals of the background function. Notably, LoRD3 has the ability to detect RDs defined by multiple real-valued dimensions, while here we focus on single-dimensional running variables. However, both LoRD3 and Porter & Yu (2015)'s approach do not explicitly consider heterogeneity in treatment assignment. To the best of our knowledge, our method is the first RD discovery algorithm that considers heterogeneity in additional covariates which affect treatment as a machine learning task.

Specifically, we take a novel approach by: 1) formulating the discovery procedure in terms of data-driven treatment assignment uptake estimation and 2) explicitly identifying heterogeneous subgroups with a higher effective sample size for both improved interpretability and statistical power. Thus, our method improves on prior work by identifying more powerful and interpretable subgroups for RD studies.

## 3. RD Discovery Framework

In the following subsections we build upon well-established econometric estimation frameworks (Imbens, 2014; Angrist & Pischke, 2008) as well as the conditional average treatment effect (CATE) estimation literature (Athey & Imbens, 2016; Chernozhukov et al., 2018) to frame our regression discontinuity discovery procedure. We then target the effective sample size of the discovered RD sample and show how optimizing for this quantity improves study feasibility by increasing statistical power.

### 3.1. Regression Discontinuity Preliminaries

Here we briefly review the potential outcomes framework for analyzing regression discontinuities (RDs); see, e.g. Imbens & Lemieux (2007); Lee & Lemieux (2009), and Cattaneo et al. (2019a) for comprehensive overviews of RDs. For an indiviual $i$, let $X_i$ be their running variable, $c$ the corresponding assignment threshold, $\vec{W}_i$ the vector of pretreatment covariates, $Z_i = \mathbf{1}[X_i \geq c]$ the threshold indicator, $Y_i$ their observed outcome, $T_i(\cdot)$ their *potential* treatment assignment, and $Y_i(\cdot)$ their potential outcome.

We note here that the potential treatment assignments are defined in terms of the threshold indicator $T_i(Z_i)$. $T_i(1)$ corresponds to the potential treatment assignment for $X_i \geq c$,

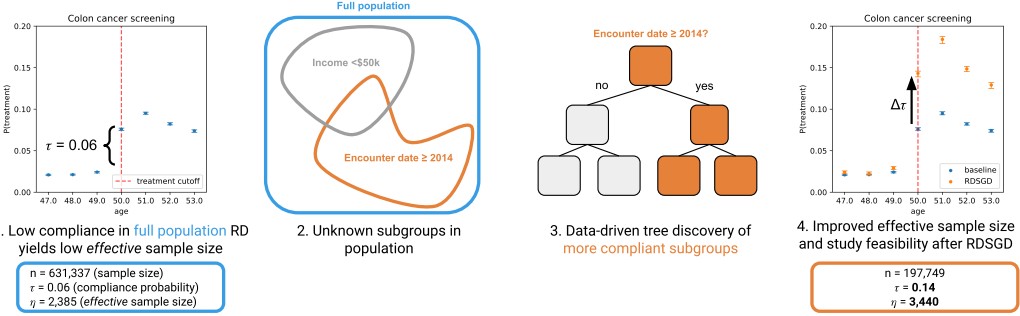

**Figure 1.** **This work develops RDSGD, a machine learning method for discovering regression discontinuity (RD) opportunities that improve study feasibility by searching for heterogeneous subgroups that have higher compliance with treatment assignment uptake (TAU).** We show one of our medical case studies (Section 6) on colon cancer screening as an illustrative example.

and $T_i(0)$ to $X_i < c$. We focus on the "fuzzy" regression discontinuity (FRD) case, which assumes that the probability of *treatment assignment uptake* jumps at threshold $c$, but not necessarily from 0 to 1 (Hahn et al., 2001).

We also define *compliers* as individuals for which $T_i(1) > T_i(0)$, namely that they receive the treatment when above the threshold, and do not receive treatment when below the threshold. Under the standard assumptions of *continuity*, *monotonicity*, and *threshold excludability* (Lee & Lemieux (2009), Appendix B.1), the treatment effect estimate $\gamma$ can be written as a ratio (Imbens & Lemieux, 2007):

$$\gamma = \frac{\lambda}{\tau} = \frac{\lim_{x\downarrow c} E[Y|X=x] - \lim_{x\uparrow c} E[Y|X=x]}{\lim_{x\downarrow c} E[T|X=x] - \lim_{x\uparrow c} E[T|X=x]} \quad (1)$$

Where $\lambda$ is the discontinuous jump in expected outcome $Y$ at the cutoff $c$, and $\tau$ is the jump in treatment assignment uptake (TAU). We turn our attention to the task of estimating $\tau$ in the context of predicting a unit's compliance status.

### 3.2. Treatment Assignment Uptake (TAU)

As the goal of our method is to identify potential RD opportunities rather than explicitly estimating downstream treatment effects, we focus on estimating treatment assignment uptake (TAU) $\hat{\tau}$, and in particular will look to maximize TAU using heterogeneity in observed covariates.

In practice, $\tau$ can be modeled using a local linear regression within a *bandwidth* $h$ around the cutoff $c$ (Hahn et al., 2001):

$$T = \tau_c Z + \beta_0 + \beta_1(1-Z)(X-c) + \beta_2 Z(X-c) + \epsilon \quad (2)$$

Where treatment assignment uptake $\tau_c$ is indexed by the cutoff $c$, $\epsilon$ is homoskedastic noise, and samples are within $X_i \in [c-h, c+h]$. We use this linear probability model estimation strategy in order to ensure causal validity, and it is commonly used in the econometric literature as an efficient approach to estimate treatment assignment (though other non-parametric methods can be used as well) (Imbens & Lemieux, 2007; Angrist & Pischke, 2008). Our RD study

discovery task can be formalized as a hypothesis test of the existence of a treatment discontinuity at threshold $c$: $H_0 : \tau_c = 0, H_A : \tau_c \neq 0$, which can be operationalized by testing the significance of the estimated $\hat{\tau}_c$ in Equation 2.

Furthermore, our estimation and subsequent maximization of treatment assignment uptake can be equivalently framed as estimation and maximization of compliance probability for subgroups at the threshold (Aronow & Carnegie, 2013; Li & Pearl, 2019; Kennedy et al., 2020) (see Appendix B.2):

**Proposition 3.1.** *For a given bandwidth $h$ and cutoff $c$, estimating $\hat{\tau}_c$ is equivalent to estimating the probability of compliance $P(T(1) > T(0))$.*

We leverage this connection in our subgroup discovery method in order maximize TAU heterogeneity to discover the most promising subgroups of data for RD analysis.

### 3.3. Heterogeneity in TAU

Beyond identifying candidate thresholds $c$ that produce significant TAU estimates, we want to find heterogeneous subgroups among our sample population at a given cutpoint $c$ in order to propose more statistically powerful RD studies. This problem can be framed as conditional compliance estimation (Aronow & Carnegie, 2013; Kennedy et al., 2020; Liu et al., 2022), where we want to identify the individuals (the compliers) to which the threshold cutoff applies, using the other pre-treatment covariates $\vec{W}$. For example, if we wanted to study the effects of breast cancer screening which is recommended at age 40 for women (Oeffinger et al., 2015), we would clearly want to exclude all men, lest their inclusion reduce any observed discontinuity in treatment due to their non-compliance with the screening guideline.

In order to identify such subgroups, we define the heterogeneous TAU estimation task. Given the standard FRD assumptions presented in Section 3.1, Kennedy et al. (2020) and Coussens & Spiess (2021) have shown that estimating the probability a unit is a complier, $\tau_c(\vec{W_i})$ (their TAU probability), can be equivalently framed as conditional average

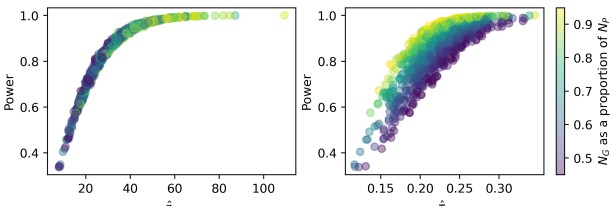

*Figure 2.* **Optimizing for effective sample size (left) increases statistical power regardless of subgroup sample size.** We simulate 1,000 randomly selected subgroups and show the relationship power has with the effective sample size $\eta_G$ (left) and treatment assignment uptake ($\tau_G$) (right), where the shading indicates different subgroup sizes as a proportion of the total population.

treatment effect (CATE) estimation (see Appendix B.3):

**Proposition 3.2.** $\tau_c(\vec{W_i})$ *for a given cutoff c can be identified as the conditional probability of compliance.*

$$P(T(1) > T(0)|\vec{W}) = \tau_c(\vec{W}) \tag{3}$$

Heterogeneous TAU $\tau_c(\vec{W})$ can thus be estimated using data-driven machine learning methods developed in the recent years for CATE estimation (e.g., Chernozhukov et al. (2018); Oprescu et al. (2019), and Padilla et al. (2021)). The machine-learned estimates of $\tau_c(\vec{W})$ will be unbiased due to the sample-splitting *honesty* property of such estimators (Athey & Imbens, 2016; Chernozhukov et al., 2018). We therefore have data-driven methods available for estimating $\tau_c(\vec{W})$ for a given RD threshold $c$.

Because our goal is to identify subgroups of individuals where treatment assignment uptake varies, we choose to use tree-based approaches (Athey & Imbens, 2016; Oprescu et al., 2019; Athey et al., 2019) for the estimation problem, which provide valid, honest, and interpretable subgroup populations defined by the learned causal tree's nodes. In particular, we can distill a tree-based model that estimates $\hat{\tau}(\vec{W})$ into a single decision tree, and extract heterogeneous subgroups that correspond to the learned tree's nodes (Athey et al., 2019; Athey & Imbens, 2016; Battocchi et al., 2019). Tree-based CATE models thus provide a data-driven approach for identifying interpretable subgroups that have heterogeneous treatment assignment uptake.

### 3.4. From TAU to effective sample size

Though we have established the TAU objective for heterogeneous treatment uptake at a threshold and an approach to identify subgroups using CATE estimators, in order to actually increase power in finite samples we cannot only account for $\tau_c(\vec{W})$; we also need to consider the sample size of the given subgroup. Solely maximizing for TAU when discovering subgroups may not yield higher power, as it is possible for such an objective to select infeasibly small subgroups with higher TAU: in our breast cancer example, a subgroup of ten women may have a higher TAU than a

subgroup of size 1,000 with 50% women, but we would much prefer the latter subgroup in terms of study feasibility.

Thus, we propose to target the *effective sample size* (Liu et al., 2022; Heng et al., 2020) of a given subgroup $G$ $\eta_G$, which explicitly accounts for both the TAU as well as the size of the subgroup. Let $P = (\vec{W_i}, Z_i, T_i, X_i)_{i=1}^{N_P}$ represent the "baseline population" samples i.e., all of the samples within the data bandwidth $h$ for a cutoff $c$ where $N_P$ is the sample size, and $G = (\vec{W_j}, Z_j, T_j, X_j)_{i=1}^{N_G}$ represent the samples that are part of a subgroup $G$, where $N_G$ is the sample size. The TAU for a subgroup $G$ is defined as $\tau_G := \tau_c(\vec{W_G})$, where $\vec{W_G}$ are the pre-treatment covariates that define a sample's membership in group $G$; similarly, $\tau_P$ is the TAU for all of the samples in the baseline population. Letting $\mathbb{E}_G[\cdot]$ be expectations taken over the subgroup $G$, the effective sample size $\eta_G$ is then:

$$\eta_G = N_G \tau_G^2 = N_G(\mathbb{E}_G[T|Z=1] - \mathbb{E}_G[T|Z=0])^2 \tag{4}$$

The intuition behind $\eta_G$ is that only units compliant with the threshold indicator $Z$ contribute to the treatment effect estimation. Furthermore, non-compliers can be seen as contributing noise to the estimate, so the "effective" sample is the nominal sample size scaled by a quantity of $\tau_G$, which is the probability of compliance with the threshold indicator (Proposition 3.1). This is a desirable quantity to maximize as it has been shown that the variance of a fuzzy RD estimator will decrease as the effective sample size increases (Liu et al., 2022; Coussens & Spiess, 2021; Heng et al., 2020). An important consequence of this relationship between effective sample size and variance is that power increases as effective sample size increases (Appendix B.4):

**Proposition 3.3.** *Statistical power is a non-decreasing function of $\eta_G$, regardless of subgroup size $G$.*

Maximizing effective sample size $\eta$ is therefore a superior objective than maximizing heterogeneous TAU alone as it is possible to select a small subgroup $G$ that has a high TAU but will ultimately still have lower power than the baseline population sample. We demonstrate this empirically in Figure 2, which in conjunction with Proposition 3.3 motivates the use of $\eta_G$ in our subgroup discovery algorithm.

### 3.5. A novel test statistic for effective sample size

Furthermore, when discovering subgroups that have higher effective sample size than the baseline population, we want to ensure that the differences are not due to noise in the selected samples. To formalize this, we develop a novel testing framework for whether the effective sample size for a subgroup $G$ is greater than that of the whole population $P$: $H_0 : \eta_G - \eta_P = 0, H_A : \eta_G - \eta_P > 0$. The corresponding test statistic is then:

$$t_\eta = \frac{\eta_G - \eta_P}{\sqrt{\text{Var}[\eta_G - \eta_P]}} \tag{5}$$

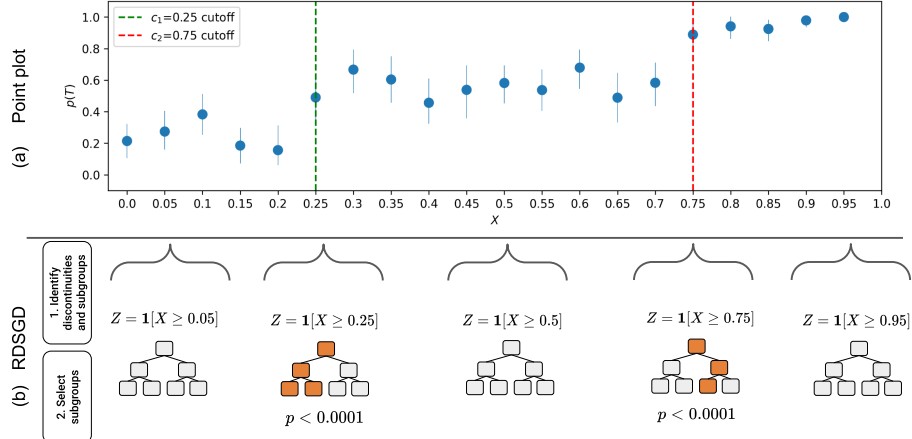

*Figure 3.* **RDs with heterogeneous cutoffs have smaller TAU (a), but RDSGD can correctly identify cutoffs with hetereogeneity (b).** Figure 2a is a pointplot of treatment probabilities across different running variables $X$ (95% CIs). 2b is a pictorial representation of RDSGD (Algorithm 1), where causal trees are fit to each candidate threshold $Z$ which generate subgroups with higher effective sample sizes (step 1) and statistically significant subgroups are selected (orange nodes, step 2).

Though $\mathrm{Var}[\eta_G - \eta_P]$ can be difficult to derive as groups $G$ and $P$ are overlapping, we leverage properties of influence functions (Newey & McFadden, 1994; Kahn, 2022) to construct a consistent estimator for this variance term (Appendix B.5). As Empirical analogues of $\eta$ (Equation 4) can be easily calculated using sample means, we are able to estimate the test statistic $t_\eta$. We verify this test statistic behaves correctly asymptotically under the null hypothesis (Figure B.1). We thus have established a valid statistical test for selecting subgroups that have a larger effective sample size, allowing us to leverage heterogeneity in treatment assignment uptake for improved study power.

## 4. Methodology

We use the mathematical framework for RD discovery and statistical testing for the effective sample size presented in Section 3 to implement our **RDSGD** (Regression Discontinuity SubGroup Discovery) method, which is outlined in Algorithm 1 and visualized in Figure 3. RDSGD comprises of two main steps: 1) identification of candidate thresholds and subgroups with higher effective sample size, and 2) subsequent selection of subgroups.

### 4.1. Identifying Discontinuities and Subgroups

In order to discover regression discontinuities with potential heterogeneity, we must first identify candidate thresholds. Given a set of cutpoints $C_X = \{c_1, c_2, ...\}$ for a running variable $X$, RDSGD analyzes thresholds $c \in C_X$. It first generates threshold indicator $Z := \mathbf{1}[X \geq c]$ and selects a bandwidth $h_c$ (Algorithm 1, steps 1a-b), which can be chosen by the user or by a data-driven selection process (Cattaneo et al., 2019a; Imbens & Kalyanaraman, 2009).

RDSGD then computes the baseline population effective

---

**Algorithm 1** RD SubGroup Discovery (RDSGD)

1. **Identify discontinuities and subgroups.**

   For $c \in C_X$:
   (a) Select bandwidth $h_c$ of analysis
   (b) Generate threshold indicator $Z := \mathbf{1}[X \geq c]$
   (c) Select baseline population $P = \{(\vec{W}_i, Z_i, T_i, X_i) \mid i \in [c - h_c, c + h_c]\}$ and compute effective sample size $\hat{\eta}_P$
   (d) Fit subgroup tree model $\hat{f}$ estimating $\hat{\tau}(\vec{W})$ (Eq. 3)
   (e) Obtain subgroups $G_{s,c} = \{(\vec{W}_i, Z_i, T_i, X_i) \mid i \in s\}$ for each node $s$ in $\hat{f}$ and corresponding subgroup effective sample size $\hat{\eta}_{G_s}$
   (f) Output subgroups with stat. sig. greater effective sample size $G_c = \{G_{s,c} \mid \hat{\eta}_{G_s} > \hat{\eta}_P\}$

2. **Select subgroups.**

   (a) For each subgroup $G_{s,c} \in \bigcup_{c \in C_X} G_c$:
       i. Select data $X_G = \{X_j \mid (j \in G_{s,c})\}$
       ii. Fit local linear estimator $\hat{T}(X_G, c)$ (Eq. 2), obtain TAU estimate $\hat{\tau}_G$ and p-value $p_{\hat{\tau}_G}$
   (b) Compute corrected significance level $\tilde{\alpha}$
   (c) Output discovered cutoffs and subgroups: $D_X = \{(c, G_{s,c}) \mid (p_{\hat{\tau}_G} < \tilde{\alpha})\}$

---

sample size $\hat{\eta}_P$ for subsequent subgroup comparison (step 1c). Because $X$ is real-valued, it can theoretically yield infinite potential cutpoints. However, in many situations (such as the clinical contexts we consider) the grid of candidate cutpoints for $X$ can be sensibly defined in terms of the running variable. For example, in whole years for age-based clinical guidelines, or at the precision of lab result readings (e.g., 0.1 increments for A1C %). In situations where the candidate cutpoints do not have an sensible definition, other data-driven methods such as LoRD3 (Herlands et al., 2018) can be used to provide $C_X$ (Section 2).

Next, RDSGD generates heterogeneous subgroups for each $c$ based on the pre-treatment covariates $\vec{W}$ by estimating $\hat{\tau}_c(\vec{W})$ for the given cutoff and bandwidth (Steps 1d-e). As discussed in Section 3.3, RDSGD uses tree-based approaches to estimate $\hat{\tau}_c(\vec{W})$ and produce candidate subgroups $G_{s,c}$ for a given cutoff $c$ for RD study evaluation. RDSGD then applies the testing framework for effective sample size (Section 3.5) to determine whether the subgroup effective sample size $\hat{\eta}_{G,s}$ is greater than the baseline population effective sample size $\hat{\eta}_P$, outputting the subgroups $G_{s,c}$ that have a statistically significant larger effective sample size than the baseline population (Step 1f).

### 4.2. Selecting subgroups

Once we have candidate heterogeneous subgroups identified, we need to select the most promising subgroups in terms of study power while preserving statistical validity. Given a subgroup $G_{s,c}$ for a cutpoint $c$, RDSGD evaluates the local linear regression for treatment assignment uptake for each subgroup to test for the discontinuous jump indicative of a potential RD study (Algorithm 1, steps 2ai-ii).

In order for the TAU test to be valid, RDSGD must account for the multiple comparisons across the set $G_X$ of *all* the subgroups considered for running variable $X$, including the candidate subgroups generated in Step 1e. RDSGD thus applies a Bonferroni correction (Wasserman, 2010) to produce the adjusted significance level $\tilde{\alpha}$ (Step 2b).

Finally, RDSGD outputs discovered subgroups and cutoffs based on $\tilde{\alpha}$ (Step 2c). By leveraging connections between 1) treatment assignment uptake estimation and machine-learned CATE estimation as well as 2) our statistical testing framework for effective sample size (Sections 3.4-3.5), RDSGD is a data-driven RD discovery procedure that directly uses potential heterogeneity among subgroups to identify more powerful RD studies (Algorithm 1, Figure 3).

## 5. Synthetic Experiments

To validate RDSGD, we first evaluate it using synthetic data where the presence of multiple discontinuities in a given running variable can be distinguished via heterogeneity in

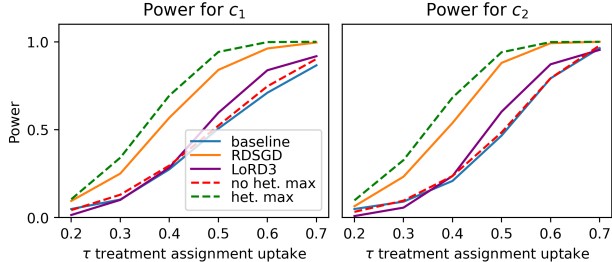

*Figure 4.* **RDSGD improves the statistical power of discovering RD opportunities by considering heterogeneity.** We simulate RDs over 500 trials for each $\tau$ and record the number of correct discoveries at $c_1$ and $c_2$ for an empirical power estimate.

*Table 1.* **RDSGD discovers more powerful subgroups over baseline in higher dimensions.** We run 500 trials with $\tau = 0.5$, recording mean power and comparing with baseline powers.

| $\dim(\vec{W})$ | $c_1$ power | $c_2$ power |
|---|---|---|
| baseline | 0.52 | 0.48 |
| 2 | **0.82**±0.15 | **0.80**±0.16 |
| 4 | **0.79**±0.15 | **0.79**±0.16 |
| 8 | **0.77**±0.15 | **0.78**±0.16 |
| 16 | **0.78**±0.15 | **0.78**±0.15 |

another observable covariate. We compare RDSGD to a baseline method that only tests the TAU regression of Equation 2 for each cutpoint $c$ (Algorithm A), and thus does not consider heterogeneity. We also make comparisons to the LoRD3 method proposed by Herlands et al. (2018).

### 5.1. Heterogeneity in One Covariate

**Data Generation.** Here we generate data where half of the units in our sample follow a fuzzy RD threshold for running variable $X \in [0, 1]$ at $c_1 = 0.25$, $Z = \mathbf{1}[X \geq c_1]$, while the other half follow a fuzzy RD threshold at $c_2 = 0.75$, $Z = \mathbf{1}[X \geq c_2]$. The threshold a particular unit follows can be identified by observed covariate $W \in [0, 1]$, with units $W < 0.50$ following threshold $c_1$ and units $W \geq 0.50$ following threshold $c_2$ (Appendix C.1-C.3). Such a scenario might arise in real-world settings where the clinical threshold varies depending on other patient attributes e.g., women with high risk of breast cancer due to hereditary factors ($W = 1$) should begin screening earlier than the recommended age of 40 for women without risk factors ($W = 0$) (Center for Disease Control, 2021). The TAUs at $c_1$ and $c_2$ will appear much smaller if covariate $W$ is not accounted for, thus this synthetic data scenario is one where we would expect RDSGD to showcase its advantages.

**Power Calculations.** In order to quantify RDSGD's performance, we need to be able to calculate the theoretical power that can be achieved for a given RD study. Given the regression framework for TAU estimation (Equation 2), we derive the theoretically achievable power levels analytically (Ap-

pendix B.8). We can then use these power calculations and our synthetic data to evaluate the baseline method, LoRD3, and RDSGD. We simulate RD datasets as described above varying across different ground-truth $\tau \in [0.2, 0.3, ..., 0.7]$, evaluating empirical power at each $\tau$ setting for correctly identifying a discontinuity at $c_1$ and $c_2$.

**Simulated results.** Our empirical results show the benefit of RDSGD (Figure 4). We calculate the theoretical power achievable without considering heterogeneity in $W$ (red dashed lines) and find that the baseline method (Algorithm A.1, blue lines) matches that power level across the different $\tau$ settings. RDSGD (Algorithm 1, orange lines) uniformly improves upon the baseline method as well as LoRD3 for both cutoffs $c_1$ and $c_2$ (Appendix C.4). Furthermore, RDSGD maintains empirical false positive rates below the nominal $\alpha = 0.05$ for every $\tau$ level due to the multiple testing corrections made (Figure C.3). Empirical power for RDSGD approaches the theoretical power when heterogeneity in $W$ is accounted for (dashed green lines). The gap between the power levels of RDSGD and the theoretical power is sensible, as in practice we lose power due to testing corrections and the data-driven tree fitting, which does not perform an exhaustive search over all subgroups.

### 5.2. Heterogeneity in Multiple Covariates

The improvement in power over baseline methods also extends to heterogeneity in multiple dimensions (Table 1), where we increase the dimensionality of the covariates $\dim(\vec{W}) \in [2, 4, 8, 16]$ that determine whether an individual complies with cutoff $c_1$ or $c_2$ and record the power of the discovered subgroups (Appendix C.5); note that we do not include LoRD3 in this comparison as one of its stated limitations is that it that does not consider heterogeneity. Though empirical power decreases slightly as the number of covariates increases (which is to be expected), RDSGD overall scales well to higher dimensions, as the average subgroup powers for both cutoffs are statistically significantly greater than the baseline theoretical powers at all $\dim(\vec{W})$ (Appendix C.5). Together, these simulation results (Figure 4, Table 1) provide empirical evidence that RDSGD can improve RD discovery in the presence of heterogeneity.

## 6. Case Study: Medical Claims Data

To evaluate RDSGD in real-world settings, we target a variety of clinical contexts where we believe RDs exist: breast cancer screening, colon cancer screening, and diabetes diagnosis. We use Optum's de-identified Clinformatics® Data Mart Database (2007-2018) for analysis, which contains claims data on, diagnoses, treatment, prescriptions, and lab results. We use all adult users with demographic data available, which span roughly 60 million unique patients over the twelve year period across the U.S. We note that each

clinical setting uses a different subset of the data as there are inclusion criteria that are specific to that setting, e.g. the presence of a lab result. Full details on demographics and the sample selection process can be found in Appendix D.

### 6.1. Data Extraction and Featurization

For each clinical setting, we target a specific running variable $X$ that corresponds to a treatment $T$ (see the first two columns of Table 2). We use LOINC, CPT, and ICD codes (McDonald et al., 2003; WHO, 2016) to identify specific laboratory results, procedures, and diagnoses. To convert longitudinal data in the claims database into tabular form, we index a patient by the first recorded presence of the running variable (Appendix D.2). We then assume a fixed window of time after the running variable is recorded for the treatment to occur, in order to account for lags in claims data reporting. If the treatment of interest appears for the patient within the fixed window, they are coded as "treated," otherwise they are "untreated." For example, in the diabetes diagnosis setting where we wish to estimate the "treatment" uptake of diabetes diagnosis, we find a patient's first recorded A1C measurement and then search the database for a subsequent type II diabetes diagnosis within the following seven days. In the screening clinical settings where age is the running variable, we use a patient's age at their first recorded preventative care visit.

Once we have both the running variable $X$ and treatment $T$ identified for a patient, we additionally query the claims database for covariates that may impact TAU heterogeneity. For all clinical settings, we consider heterogeneity across patient demographics (age, gender, race), socio-economic status (education level, household income) and claims-specific features (recorded initial encounter date, insurance type). Our data extraction method provides a pipeline for converting raw claims data into feature matrices for RDSGD.

We use the EconML package (Battocchi et al., 2019) to estimate the heterogeneous TAU tree model needed for RDSGD (Algorithm 1, step 1d). Note that due to our larger sample size, most candidate RDs that RDSGD returns will have

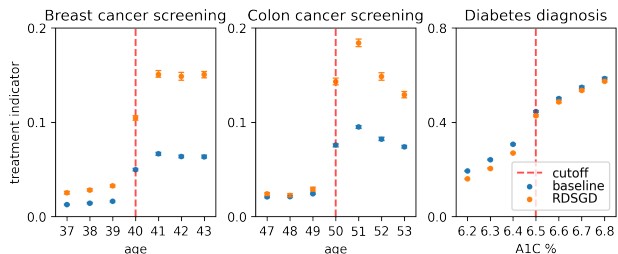

*Figure 5.* **RDSGD discovers subgroups that improve the TAU in different clinical settings.** We show the discovered cutoff for each clinical setting and the probability of treatment for the entire sample (blue) and for subgroup discovered by RDSGD (orange).

*Table 2.* **Discovered RD thresholds and subgroups in medical claims data.** We report $\hat{\tau}$ and $\hat{\eta}$ (higher is better in both cases) for baseline RDs and discovered subgroups.

| Clinical guideline | Running variable | Threshold | Subgroup discovered | Baseline $\hat{\tau}$ (SE) | RDSGD $\hat{\tau}$ (SE) | Baseline $\hat{\eta}$ | RDSGD $\hat{\eta}$ |
|---|---|---|---|---|---|---|---|
| Breast cancer screening | Age | $\geq 40$ | Gender = Female | 0.039 (0.0009) | **0.081 (0.002)** | 970.1 | **2194.4** |
| Colon cancer screening | Age | $\geq 50$ | Encounter date > 2014-05-05 | 0.059 (0.001) | **0.13 (0.002)** | 2385.5 | **3440.4** |
| Type 2 diabetes diagnosis | A1C % | $\geq 6.5$ | Encounter date > 2010-05-11 | 0.089 (0.003) | **0.11 (0.003)** | 3694.0 | **4730.1** |

power approaching 1, so here we compare the estimated TAU and effective sample size $\eta$ of the discovered subgroups to the baseline RD study without considering heterogeneity.

### 6.2. Results

The most promising discovered RD thresholds and subgroups for each clinical setting are shown in Table 2, with treatment probability point plots shown in Figure 5.

**Breast Cancer.** As our validation case, RDSGD correctly identifies that breast cancer screening only applies to women at a screening age of 40 (Oeffinger et al., 2015), doubling the effective sample size from 970.1 to 2194.4.

**Colon Cancer.** RDSGD correctly identifies the recommended screening age of 50 for colon cancer, and additionally discovered a subgroup of patients who were more likely to be screened at the threshold; these individuals had an encounter date later than 2014-05-05, producing a subgroup with a higher effective sample size than the baseline population (2385.5 vs. 3440.4). This could be due to an increase of adherence to screening resulting from a guideline update that occurred approximately in the same time period (US Preventive Services Task Force, 2016).

**Type 2 Diabetes.** RDSGD identifies the A1C cutoff of 6.5% for diabetes diagnosis (American Diabetes Association, 2010), and also identifies a subgroup of patients more likely to be compliant with the cutoff, increasing the effective sample size over baseline from 3694.0 to 4730.1. This subgroup excludes patients who have encounter dates before 2010-05-11, which aligns with intuition as A1C was not introduced as a diagnostic criteria for diabetes until 2010.

### 7. Discussion

In this paper we have proposed RDSGD, a method for data-driven regression discontinuity (RD) discovery that produces interpretable subgroups by optimizing for the *effective sample size* through a causal machine learning framework. We demonstrate through synthetic studies how RDSGD provides power improvements in the presence of heterogeneity.

We apply RDSGD to a variety of clinical settings that both validate the correctness of our method as well as discover new RDs for practitioners to investigate. We now discuss how our method fits into clinical study use cases as well as highlight limitations and future work opportunities.

### 7.1. Clinical Use Cases via Interpretable Subgroups

RDSGD is most useful in scenarios for treatment effect estimation when explicit randomization of the treatment is not possible. While our method only discovers RD opportunities within the data (the so-called treatment assignment regression, Section 3.1) and does not make treatment *effect* estimates, the primary goal is to identify quasi-experimental randomness that can be used to estimate the downstream effects of the given treatment $T$ on an outcome of interest $Y$. For example, in our A1C diabetes case study, a practitioner may wish to study the effect of the A1C cutoff on different outcomes, such as metformin prescription, follow-up A1C levels, or heart attack incidence. By identifying both the cutoff as well as subgroups where the effective sample size is stronger, any downstream treatment effect estimation a practitioner wishes to conduct has both: 1) an identified variable that provides quasi-experimental randomness and 2) an interpretable cohort to which it applies.

Furthermore, RDSGD could be used to investigate implicit differences in treatment assignment. Because the subgroups produced by RDSGD define clear inclusion criteria based on the path of the fitted causal tree, it can be used to identify sources of bias in treatment decisions such as those documented in FitzGerald & Hurst (2017); Hausmann et al. (2013); Hoffman et al. (2016).

### 7.2. Limitations and Future Work

We highlight some limitations and opportunities for future work. First, we note that there are issues of causal validity that still need be addressed when applying RDSGD to real data. As discussed above, we do not make treatment effect estimates as part of our method and defer that step to practitioners, who are free to choose which outcome $Y$ they wish

to study. However, care must be taken when moving forward to the treatment effect estimation stage (estimating $\gamma$, Section 3.1). When making treatment effect estimates with identified cutoffs we need to be mindful of the *exclusion restriction* assumption (Imbens & Lemieux, 2007), which can be violated when the cutoff decision $Z$ influences the outcome $Y$ outside of its influence on $T$. In our case studies, this assumption can be somewhat difficult to justify when using insurance claims data as we only have partial visibility into how patients interact with their healthcare provider.

Moreover, there are additional limitations in using medical claims data, as there may be selection biases in healthcare utilization as well as potential under-reporting of diagnoses and treatments of interest (van Walraven & Austin, 2012; Jensen et al., 2015). Thus, when moving forward with RD studies that use cutoffs identified by RDSGD, practitioners need to work in close concert with domain experts to ensure that causal validity is maintained. Validation tests specific to RDs, such as whether the running variable has been manipulated, also need to be run to ensure a plausible design (McCrary, 2008; Imbens & Lemieux, 2007).

We also note that because RDSGD uses tree-based methods to identify heterogeneous TAU subgroups, it is inherently greedy (see Figure 4 where RDSGD approaches, but does not achieve max power). Though our use of trees was a deliberate design decision made to maintain interpretability and scalabililty to large datasets (Wu et al., 2022), future work could investigate other methods that are optimal in terms of TAU maximization: for example, applying policy learning methods that maximize power in randomized trials to RDs (Spiess & Syrgkanis, 2021). Future work could also investigate using other CATE estimators to identify subgroups that maximize effective sample size.

Additionally, further analysis of false positive control can improve our method. We conservatively perform a Bonferroni correction, but due to the correlated nature of candidate RDs across the same running variable, less stringent corrections could be used to maintain the nominal significance levels while further improving power (e.g., Hu et al. (2010)).

### 7.3. Conclusion

Here we have introduced a machine learning-based method for RD study discovery, RDSGD, which identifies interpretable subgroups that increase the effective sample size of a study. RDSGD is shown to be effective in both simulated and real data settings, and could provide new avenues for more credible causal inference studies in medicine through quasi-experimental designs.

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

## A. Baseline algorithm details

We give the full baseline RD discovery procedure used for comparison with RDSGD in Algorithm A.1.

---

**Algorithm A.1** Baseline method: RD threshold discovery

---

1. For $c \in C_X$:

    (a) Select bandwidth $h_c$ for treatment regression (Section 3.1)

    (b) Select data $X = \{X_i \mid X_i \in [c - h_c, c + h_c]\}$

    (c) Fit estimator $\hat{T}(X, c)$ of Equation 2 to obtain TAU estimate $\hat{\tau}_c$ and output p-value $p_{\hat{\tau}_c}$

2. Compute corrected corrected significance level $\tilde{\alpha} = \alpha/|C_X|$

3. Output discovered cutoffs and bandwidths: $D_X = \{(c, h_c) \mid p_{\hat{\tau}_c} < \tilde{\alpha}\}$

---

## B. Mathematical details

### B.1. FRD and IV assumptions

We assume the following standard FRD assumptions for valid estimation (Lee & Lemieux, 2009) to identify Equation **??**:

- **Continuity**. Both the potential outcomes $Y(1)$ and $Y(0)$ are continuous as a function of the running variable $X$:

$$E[Y_i(1)|X], E[Y_i(0)|X] \text{ continuous over domain of } X \tag{6}$$

- **Monotonicity**. $X$ crossing the cutoff cannot simultaneously cause some units to take up and others to reject the treatment (also known as the "no defiers" assumption):

$$T(1) \geq T(0) \tag{7}$$

- **Excludability of crossing the threshold.** $X$ crossing the cutoff cannot impact $Y$ except through impacting the treatment:

$$Y(T = t, Z = z) = Y(T = t) \tag{8}$$

    Where $Y(T = t, Z = z)$ is the potential outcome $Y$ that would have been observed if both $T = t$ and $Z = z$.

Given the equivalence between FRDs and IVs we discuss in the main text (Section 3.1), many of the assumptions needed for valid TSLS estimation are equivalent (Imbens, 2014; Kennedy et al., 2020):

- **Consistency**. If $Z = z$ and $T = t$, then the observed outcomes of $T$ and $Y$ are the potential outcomes under $Z = z$ and $T = t$.

$$T = ZT(1) + (1 - Z)T(0) \tag{9}$$
$$Y = TY(1) + (1 - T)Y(0) \tag{10}$$

- **Unconfounded instrument**. The instrument is unconfounded with the potential treatment given the observed covariates.

$$Z \perp T(1), T(0) \mid \vec{W} \tag{11}$$

- **Monotonicity**. The instrument cannot simultaneously cause some units to take up and others to reject the treatment.

$$T(1) \geq T(0) \tag{12}$$

- **Exclusion restriction**. The instrument cannot impact $Y$ except through impacting the treatment:

$$Y(T = t, Z = z) = Y(T = t) \tag{13}$$

### B.2. Proposition 3.1 details: compliance and TAU equivalence

Imbens & Lemieux (2007); Hahn et al. (2001) show how a fuzzy regression discontinuity estimated via local linear regression for a given cutoff $c$ and fixed bandwidth $h$ is numerically to a two-stage least squares (TSLS) estimation problem with the following additional regressors:

$$\begin{pmatrix} 1 \\ \mathbf{1}[X_i < c](X_i - c) \\ \mathbf{1}[X_i \geq c](X_i - c) \end{pmatrix} \tag{14}$$

Note that the instrument $Z_i$ is defined as $Z_i = \mathbf{1}[X_i \geq c]$, the same as our RD cutoff indicator. These regressors thus give the form of the treatment regression in Equation 2.

Imbens & Rubin (2015) further show that under IV assumptions of consistency, unconfounded instrument and monotonicity (Appendix B.1), the IV estimate of treatment effects can be expressed as:

$$\gamma_{IV} = \frac{E[Y_i(1) - Y_i(0)|\text{unit i is a complier}] \cdot \pi_{\text{comply}}}{\pi_{\text{comply}}} \tag{15}$$

Where $P(T(1) > T(0)) = \pi_{\text{comply}}$ is the probability of compliance. Within the data bandwidth of analysis $h$, we can equivalently write $\tau$ of Equation 1 in terms of $Z$ (Hahn et al., 2001):

$$\begin{aligned} \tau &= E[T|Z = 1] - E[T|Z = 0] \\ &= P(T = 1|Z = 1) - P(T = 1|Z = 0) \\ &= 1 - P(T = 0|Z = 1) - P(T = 1|Z = 0) \\ &= 1 - \pi_{\text{never taker}} - \pi_{\text{always taker}}, \text{ (unconfounded instrument)} \\ &= \pi_{\text{comply}}, \text{ (monotonicity)} \end{aligned} \tag{16}$$

Where $\pi_{\text{always taker}}$ and $\pi_{\text{never taker}}$ are the proportions of always-takers and never-takers. We then have that $\tau$ is equivalent to the probability of compliance, and that the first stage (denominator of $\gamma_{IV}$ from Equation 15) regression of the TSLS framework estimates the probability of compliance. Thus we can use the TSLS framework for our analysis and estimation of RD TAU.

### B.3. Proposition 3.2 details: conditional compliance identification

Identification of the conditional probability of compliance follows a similar argument as Appendix B.2. Given the cutoff choice $c$ generating $Z = \mathbf{1}[X \geq c]$ and a fixed bandwidth $h$, we can use the equivalent analysis of $Z$ as an IV like we do in Appendix B.2. The conditional probability of compliance is given by:

$$P\left(T(1) > T(0)|\vec{W}\right)$$

From the monotonicity assumption, there are no defiers (units where $T(1) < T(0)$). We can thus write:

$$\begin{aligned} &P(T(1) > T(0)|\vec{W}) \\ &= 1 - P\left(T(1) = 1, T(0) = 1|\vec{W}\right) - P\left(T(1) = 0, T(0) = 0|\vec{W}\right) \end{aligned} \tag{17}$$

The latter two terms are the probability of always-takers and never-takers given covariates $\vec{W}$, respectively. From the unconfounded instrument assumption, these quantities can be *identified* (can be converted from causal quantities to estimable statistical quantities) (Imbens & Rubin, 2015):

$$P(T(1) = 1, T(0) = 1|\vec{W}) = P(T = 1|Z = 0, \vec{W})$$
$$P(T(1) = 0, T(0) = 0|\vec{W}) = P(T = 0|Z = 1, \vec{W})$$

This then gives:

$$
\begin{aligned}
P(T(1) > T(0)|\vec{W}) &= 1 - P(T = 1|Z = 0, \vec{W}) - P(T = 0|Z = 1, \vec{W}) \\
&= 1 - P(T = 1|Z = 0, \vec{W}) - (1 - P(T = 1|Z = 1, \vec{W})) \\
&= P(T = 1|Z = 1, \vec{W}) - P(T = 1|Z = 0, \vec{W})
\end{aligned}
\tag{18}
$$

Allowing us to identify the conditional probability of compliance, and thus the conditional TAU $\tau_c(\vec{W})$ as desired.

**Equivalence of CATE and heterogeneity treatment assignment uptake.** Next, we make a connection between the conditional TAU and conditional average treatment estimation (CATE).

The CATE of a treatment $T$ on outcome $Y$ is given by:

$$\text{CATE} = E[Y|T = 1, \vec{W}] - E[Y|T = 0, \vec{W}] \tag{19}$$

Given the two-stage process of RD treatment effect estimation, we not only have potential outcomes $Y(\cdot)$ but also potential treatments $T(\cdot)$. We can thus equivalently analyze the "treatment effect" the cutoff indicator $Z$ has on "outcome" $T$, yielding Equation 3. The same standard fuzzy RD assumptions that enable estimation of the treatment effect $\gamma$ at cutoff $c$ (Section 3.1) also enable estimation of the heterogeneous TAU $\tau_c(\vec{W})$ through CATE estimation frameworks (Kennedy et al., 2020; Coussens & Spiess, 2021).

### B.4. Proposition 3.3 details: power and effective sample size

We describe the relationship between $\eta$ and power. From Cattaneo et al. (2019b), we have the following power function for an $\alpha$-level two-sided test for fixed TAU $\tau$ given a local linear treatment regression:

$$\beta(\tau) = 1 + \Phi\left(\frac{\tau}{\sqrt{V}} - z_{\alpha/2}\right) - \Phi\left(\frac{\tau}{\sqrt{V}} + z_{\alpha/2}\right) \tag{20}$$

where $\Phi$ is the Normal distribution CDF, $z_t$ is its $t$th percentile (e.g. $z_a = \Phi^{-1}(a)$), and $V$ is the variance of $\tau$. Next, leveraging the equivalence between instrumental variable (IV) analysis and fuzzy regression discontinuities (FRD) established in Proposition 3.1, the variance of an FRD estimator under constant treatment effects has been shown to be (Liu et al., 2022; Coussens & Spiess, 2021):

$$V = \frac{\text{Var}[Y|Z, \text{ compliers}]}{\eta E[Z](1 - E[Z])} \tag{21}$$

Thus, variance decreases as $\eta$ increases. We note that even under the relaxation of the constant treatment effects assumption, Heng et al. (2020) and Baiocchi et al. (2014) have shown that the IV variance with $n$ samples is at least as large as the variance with $\eta$ samples of known compliers. Thus, the two-sided power function is non-decreasing as $V$ decreases, and hence when $\eta$ increases. Freeman et al. (2013) also equivalently show this relationship between IV power and $\eta$ in their power calculation analysis of Mendelian randomization studies.

Note that our statements are of the power non-decreasing as a function of $\tau$ because power is bounded ($\beta(\tau) \in [0, 1]$). We show empirically in Figure 2 that power in practice increases as $\hat{\eta}$ increases, regardless of the size of the subgroup. Data simulated in Figure B.1 follow the TAU regression data generation described in Appendix C.1 with $n = 1000$, $\tau = 0.2$, and a bandwidth of 0.5. Random subgroups of sizes uniformly distributed between 450 and 950 are drawn to show the relationship between $\tau_G$ and $\eta_G$ and power across varying $G$ sizes.

### B.5. Effective sample size test statistic derivation

From Section 3.5, in order to construct a test statistic for Equation 5 we need a consistent estimator of $\text{Var}[\eta_G - \eta_P]$:

$$\text{Var}[\eta_G - \eta_P] = \text{Var}[\eta_G] + \text{Var}[\eta_P] - 2\text{Cov}[\eta_G, \eta_P] \tag{22}$$

We use *influence functions* to empirically estimate this variance term (Newey & McFadden, 1994; Kahn, 2022). The influence function $\psi_{G,i}$ for $i$ in subgroup $G$ is given in Equation 23(see Appendix B.6 for a full derivation):

$$\psi_{G,i} = 2N_G\hat{\tau}_G\Big( \mathbf{1}[Z_i = 1]\frac{N_G}{N_{1G}}(T_i - \overline{T}_{1G}) \tag{23}$$
$$-\mathbf{1}[Z_i = 0]\frac{N_G}{N_{0G}}(T_i - \overline{T}_{0G})\Big)$$

Where $\mathbf{1}[\cdot]$ is the indicator function and $N_{zG}$ is the number of samples in subgroup $G$ where $Z = z$. This function consists of products and differences of empirical means over different subgroups which are straightforward and fast to compute. Following the properties of influence functions (Newey & McFadden, 1994; Erickson & Whited, 2002), we can next derive the variance-covariance matrix of $\eta_P, \eta_G$ as follows. Let

$$\Psi = \left[\vec{\psi}_P, \vec{\psi}_G\right]_{N_P \times 2} \tag{24}$$

where $\vec{\psi}_G$ is a vector of length $N_P$ with values at the $i$th index of $\psi_{G,i}$ if $i \in G$ and 0 otherwise. We can then compute the variance-covariance matrix of $\eta_P, \eta_G$ as follows (Appendix B.7):

$$V = \frac{1}{N_P^2}\left(\Psi^T\Psi\right) \tag{25}$$

The elements of $V$ give us empirical, consistent estimates of $\text{Var}[\eta_G]$, $\text{Var}[\eta_P]$, and $\text{Cov}[\eta_G, \eta_P]$ due to the properties of influence functions, allowing us to calculate $t_\eta$ (Equation 5).

### B.6. Effective sample size influence function

We first give the influence function of a sample $i$ on $\tau_G$, which can be seen as a difference-in-means estimator (Imbens & Rubin, 2015) as shown by Kahn (2022):

$$\psi_{\tau_G,i} = \mathbf{1}[Z_i = 1]\frac{N_G}{N_{1G}}(T_i - \overline{T}_{1G}) - \mathbf{1}[Z_i = 0]\frac{N_G}{N_{0G}}(T_i - \overline{T}_{0G}) \tag{26}$$

We can then apply the influence function chain rule (Kahn, 2022) to obtain the influence function for the effective sample size. For an estimator $\hat{\theta}$ such that $\hat{\theta} = T(\hat{\theta}_j, ..., \hat{\theta}_n)$, the influence function of $\hat{\theta}$ is:

$$\psi_{\hat{\theta},i} = \sum_j^n \frac{\partial T}{\partial \hat{\theta}_i}\psi_{\hat{\theta}_j,i} \tag{27}$$

The influence function $\psi_{\eta_G,i}$ of $\eta_G$ for a given sample in subgroup $G$ is thus:

$$\psi_{\eta_G,i} = \frac{\partial}{\partial \hat{\tau}_G}N_G\hat{\tau}_G^2$$
$$= 2N_G\hat{\tau}_G\psi_{\tau_G,i}$$
$$= 2N_G\hat{\tau}_G\left(\mathbf{1}[Z_i = 1]\frac{N_G}{N_{1G}}(T_i - \overline{T}_{1G}) - \mathbf{1}[Z_i = 0]\frac{N_G}{N_{0G}}(T_i - \overline{T}_{0G})\right) \tag{28}$$

Note that in the above text we refer to $\psi_{\eta_G,i}$ as $\psi_{G,i}$ to reduce notational clutter.

## B.7. Effective sample size variance-covariance matrix

We follow Kahn (2022) and Erickson & Whited (2002) for the derivation of the variance-covariance matrix between $\eta_G$ and $\eta_P$ in Equation 25. The distribution of an estimator $\theta$ is equivalent to $\frac{1}{\sqrt{N}} \sum_i^N \psi_{\theta,i}$. Erickson & Whited (2002) show:

$$\sqrt{N_P} \begin{pmatrix} \hat{\eta}_P - \eta_P \\ \hat{\eta}_G - \eta_G \end{pmatrix} = \frac{1}{\sqrt{N_P}} \sum_i^{N_P} \begin{pmatrix} \psi_{\eta_P,i} \\ \psi_{\eta_G,i} \end{pmatrix} \xrightarrow{d} \mathcal{N}\left( \begin{pmatrix} 0 \\ 0 \end{pmatrix}, \mathbb{E} \begin{pmatrix} \psi_{\eta_P,i}^2 & \psi_{\eta_P,i}\psi_{\eta_G,i} \\ \psi_{\eta_P,i}\psi_{\eta_G,i} & \psi_{\eta_G,i}^2 \end{pmatrix} \right) \tag{29}$$

Kahn (2022) then shows that the variance-covariance matrix defined in Equation 25 gives us the variances of $\eta_G, \eta_P$ on the diagonal and the covariance of $\eta_G$ and $\eta_P$ on the off-diagonal, allowing us to compute the test statistic in Equation 5. As a sanity check, we verify that the distribution of the computed test statistic under the null hypothesis behaves correctly in empirical simulations (Figure B.1). Data simulated in Figure B.1 follow the TAU regression data generation described in Appendix C.1 with $n = 200$, $\tau = 0.5$, and a bandwidth of 0.5. Random overlapping subgroups of size 100 are drawn to test the null distribution.

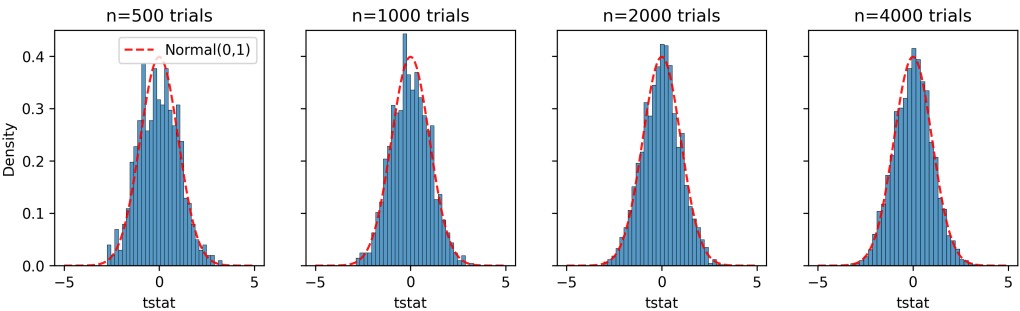

*Figure B.1.* Our overlapping $\eta$ hypothesis test produces well-behaved null distributions, yielding valid p-values. We conduct six different simulations with varying numbers of trials under the null hypothesis that the overlapping groups have the same effective sample size.

## B.8. Closed-form power calculations

We use Equation 20 shown in Appendix B.8 to compute the theoretical power of a treatment regression. To calculate $V$, Imbens & Lemieux (2007) give a closed form solution for the asymptotic variance of the treatment regression, assuming a symmetric bandwidth $h$:

$$V = \frac{8 \cdot p_{\text{bw}}}{n}(\sigma_{T,l}^2 + \sigma_{T,u}^2) \tag{30}$$

where $n$ is the total sample size, $\sigma_{T,l}^2$ is the TAU variance below the cutoff, $\sigma_{T,u}^2$ is the TAU variance above the cutoff, and $p_{\text{bw}}$ is the fraction of units in the sample that are included in the bandwidth $h$.

Since $T$ is binary, we have that:

$$\sigma_{T,l}^2 = \lim_{x \uparrow c} Var(T|X = x) = \mu_{T,l} \cdot (1 - \mu_{T,l})$$
$$\sigma_{T,u}^2 = \lim_{x \downarrow c} Var(T|X = x) = \mu_{T,u} \cdot (1 - \mu_{T,u})$$

where:

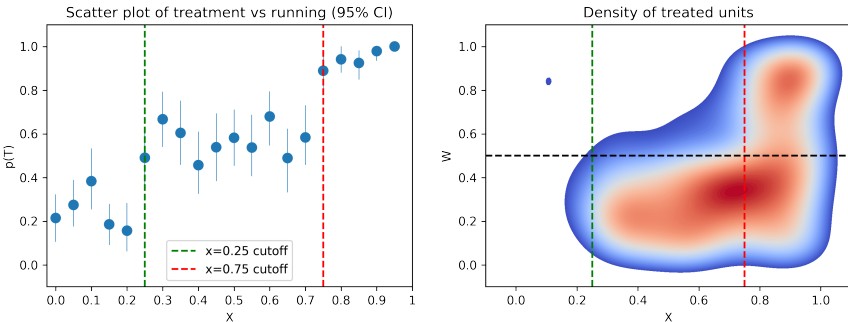

*Figure C.2.* **Heterogeneity in TAU can be observed when considering the additional covariate** $W$**.** Left is a pointplot of treatment assignment uptake probabilities as a function of the running variable $X$ (the same as pictured in Figure 3) for a single trial of our simulation, with error bars as 95% CIs and a nominal $\tau = 0.7$. Right is a density plot of treated units as a function of both $X$ and $W$.

$$\mu_{T,l} = \lim_{x \uparrow c} Pr(T = 1 | X = x)$$

$$\mu_{T,u} = \lim_{x \downarrow c} Pr(T = 1 | X = x)$$

$\mu_{T,u}$ and $\mu_{T,l}$ can be computed given our simulated data generating process, giving us a closed form solution of the theoretical power that can be achieved in our synthetic experiments.

## C. Synthetic experiment details

### C.1. TAU regression setup

For our blended RD simulation scenario, data are generated in the following manner. Let $X_i \sim \text{Unif}(0, 1)$ be the running variable for unit $i$. We generate threshold indicator $Z_i = \mathbf{1}[X_i > c]$, where $c$ is the chosen treatment threshold. Each unit's probability of treatment assignment is defined as:

$$p_i = \tau Z_i + \nu X_i + \eta + \psi_i \tag{31}$$

where $\tau$ is the true TAU, $\nu$ the coefficient determining the running variable's effect on the outcome, $\eta$ a constant, and $\psi_i$ a Gaussian noise term. For each generated data set, we vary $\tau$ and draw $\nu \sim N(0, 0.1)$ and set $\eta = 0.2, \nu = 0.05$. $p_i$ values are clamped to $[0, 1]$, with an individual's treatment assignment then defined as $T_i \sim \text{Bern}(p_i)$.

### C.2. Blended RD in one covariate

For the simulation setting shown in Section 5, for each unit we additionally draw $W_i \sim \text{Unif}(0, 1)$, with the cutoff $c$ a unit complies with being determined by:

$$c = \begin{cases} 0.25 \text{ if } W_i < 0.5 \\ 0.75 \text{ if } W_i \geq 0.5 \end{cases} \tag{32}$$

For each trial of our synthetic experiment, we draw 1000 units with half of the units complying with the lower cutoff and half complying with the upper cutoff. Differences in TAU can be observed when visualizing heterogeneity in $W_i$ (Figure C.2).

### C.3. Treatment assignment uptake reduction in the presence of heterogeneity

As our simulated data contains two equally-sized populations that comply with different cutoffs, the observed $\tau$ will effectively be reduced by half, since for each cutoff $c_1, c_2$, half of the units do not comply with the jump in TAU at that

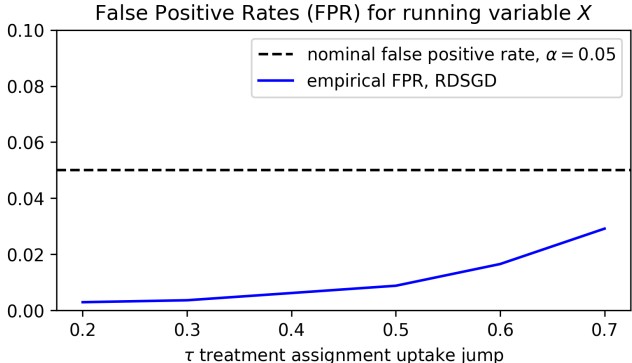

*Figure C.3.* **RDSGD maintains empirical false positive rates below the nominal $\alpha$ level.** We simulate RDs with multiple running variables over 500 trials for each $\tau$ and record the number of false threshold discoveries.

point. For example, at a given $\tau$ and $n$, we can estimate the observed TAU at $c_1$ for our sample by computing $\mu_{T,u} - \mu_{T,l}$:

$$\mu_{T,u} = \lim_{x \downarrow c_1} Pr(T = 1 | X = x)$$
$$= \frac{\nu \cdot c_1 \cdot n + \eta \cdot n + (\tau \cdot \frac{n}{2})}{n} = \nu c_1 + \eta + \frac{1}{2}\tau$$

$$\mu_{T,l} = \lim_{x \uparrow c_1} Pr(T = 1 | X = x)$$
$$= \frac{\nu \cdot c_1 \cdot n + \eta \cdot n}{n} = \nu c_1 + \eta$$

Thus we have $\mu_{T,u} - \mu_{T,l} = \frac{1}{2}\tau$.

## C.4. Heterogeneity in one covariate: simulation details

Results presented in Section 5.1 use data generated according to Appendix C.1-C.2. We use a fixed bandwidth $h = 0.25$ to ensure that the closed form oracle powers calculated in Appendix B.8 are valid. Causal forests were fit according to default parameters specified in the EconML package (Battocchi et al., 2019) (with *honesty* enabled for valid and unbiased inference), and a fixed depth of 3 and minimum leaf size of 100 were used for subsequent CATE causal trees distilled from the forests to ensure subgroups remained interpretable. We note that the causal forest implementation in EconML by default runs a two-fold cross validation internally when selecting hyperparameters for the LogisticRegressionCV scikit-learn models for treatment, which searches over L2 regularization parameters in a grid of 10 values between $1e-4$ and $1e4$ using the default accuracy criterion. Seeds were passed to machine learning models to ensure reproducibility. All 500 trials were seeded with their trial number, and once implementation was complete experiments were run twice to validate reproducibility. We run Herlands et al. (2018)'s RD discovery method, LoRD3, according to recommended parameters in their code repository, setting (in their notation) $k = 100$ and $z = \{X, W\}$ so that information from both $X$ and $W$ are used. All simulations were run on a Ubuntu 20.04 LTS server, with a 24-core Intel i9-7920X CPU and 94 GB RAM.

The false positive rates shown in Figure C.3 are computed based on the number of significant discontinuities discovered that do not equal $c_1$ or $c_2$ divided by the total number of tests over the 500 trials for each $\tau$ level. As our empirical power metric amounts to a count of "successful" detections of statistically significant discontinuities at $c_1$ and $c_2$ out of the 500 trials, we use a $\chi^2$ test (or corresponding Fisher's exact test if count numbers are not sufficient) to compare RDSGD's performance with the baseline discovery method. With the exception of the $\tau = 0.2, c_2$ case, all the differences between RDSGD and the baseline algorithm are statistically significant with $p < 0.001$. We also show comparisons to LoRD3 in Figure **??**, where RDSGD outperforms it across all $\tau$ levels; this is to be expected due to the stated limitations of LoRD3 with regards to heterogeneity.

## C.5. Heterogeneity in multiple covariates: simulation details

Results presented in Section 5.2 used data generated according to Appendix C.1. Instead of having a single covariate govern the choice of $c$ as in Appendix C.2, we generate $\dim(\vec{W})$ covariates $W_1, W_2, ...$ for each unit $i$ using scikit-learn's make_regression() method, producing output $\omega_i$. We then scale $\omega_i$ to fall in the range $[0, 1]$ with sample mean 0.5, and determine which cutoff a unit complies with by:

$$c = \begin{cases} 0.25 \text{ if } \omega_i < 0.5 \\ 0.75 \text{ if } \omega_i \geq 0.5 \end{cases} \tag{33}$$

We fix ground truth $\tau = 0.5$ for the simulation trials and calculate the baseline oracle power according to Appendix B.8. All hyperparameters, hardware, and seeding strategy for the 500 trials are the same as described in Appendix C.4.

All statistically significant subgroups discovered by RDSGD are recorded, and their power is computed according to Equation 20. Means and standard deviations reported in Table 1 are taken across all trials. For significance testing, we use a one-sample t-test comparing the subgroups in each cell of Table 1 with their corresponding baseline oracle powers. All tests were statistically significant at $p < 0.001$.

# D. Clinical setting and cohort details

## D.1. Justifying use of private claims dataset

In order to evaluate RDSGD in a real-world setting where our clinical collaborators can help verify discovered candidate RD studies, we needed a large-scale clinical data source that spanned general healthcare settings with enough data granularity on individuals so our method can leverage potential TAU heterogeneity across common demographic covariates (described in Appendix D.2). The claims dataset that we use has the advantage of having an array of disease classes and visibility into patient information in general healthcare settings, as opposed to publicly available datasets such as MIMIC which focuses on a very specific context (critical care). Working with such detailed patient information necessitates adherence to federal HIPAA privacy rules concerning privacy, which restrict access to "protectable health information"; we do however provide descriptive statistics of the cohort presented in Table D.1.

## D.2. Data extraction per clinical setting

**Breast cancer screening**. We extract a patient's first recorded routine preventative care visit as designated by ICD and CPT codes. The treatment indicator $T$ for a patient is whether they received a breast cancer screen as designated by ICD code within 7 days of the recorded encounter date. The running variable $X$ is the patient's age at the initial encounter date (note that in order to protect patient privacy, the claims database only has resolution to a patient's year of birth). We consider candidate thresholds of $C_X = [40, 45, ..., 60]$ with the data-driven bandwidth selected to be 4, at age increments of 5 years to align with typical screening guideline values.

**Colon cancer screening**. Similar to the breast cancer setting, we use a patient's first recorded routine preventative care visit. The treatment indicator $T$ for a patient is whether they received a colon cancer screen within 7 days of the recorded encounter date, and age is the running variable $X$ with candidate thresholds of $C_X = [40, 45, ..., 60]$ and bandwidth 4, at age increments of 5 years to align with typical screening guideline values.

**Type 2 diabetes diagnosis**. We extract patient's first recorded A1C measure as designated by LOINC codes and use it as the running variable $X$. The treatment indicator $T$ for a patient is whether a type II diabetes diagnosis ICD code appears in their record within 30 days of the first recorded A1C measure. We consider candidate thresholds of $C_X = [5.0, 5.1, 5.2, ..., 7.5]$ and a data-driven bandwidth selected to be 0.4 as this is the standard range of A1C values, with the lab readings having precision to one decimal place.

In all three clinical settings, we exclude patients that have a recorded treatment indicator code prior to their initial encounter date, as well as patients that do not have recorded demographic information. The following covariates are included as $\vec{W}$ for each patient (unordered categorical variables are one-hot encoded, while ordinal variables are coded as integers): gender, encounter date, insurance type (Medicare vs. commercial), race, education level, household income range.

## D.3. Full result and cohort details

Claims data analyses were run on a secure CentOS Linux 7 server with a 40-core Intel Xeon E5-4650 CPU and 504 GB RAM. We follow the same hyperparameter and model training strategy as described in Appendix C.4. For the RD candidate cutoff identified in each clinical setting, we show the best subgroup in terms of effective sample size in Table 2. We describe the demographics of patients within the RD bandwidth of analysis in Table D.1.

*Table D.1.* **Demographic details for each clinical setting within RD bandwidth.**

|  | Breast cancer screen, age ≥ 40 | Colon cancer screen, age ≥ 50 | Type 2 diabetes diagnosis, A1C ≥ 6.5 |
|---|---|---|---|
| Sample size | 631,337 | 691,559 | 389,257 |
| Mean age (SD) | 39.9 (1.97) | 49.9 (1.98) | 60.6 (13.4) |
| Gender (%) |  |  |  |
|     Male | 338,052 (53.5) | 342,930 (49.6) | 181,137 (46.5) |
|     Female | 293,285 (46.5) | 348,629 (50.4) | 208,120 (53.5) |
| Race (%) |  |  |  |
|     White | 425,331 (67.4) | 513,856 (74.3) | 226,714 (58.2) |
|     Black | 65,453 (10.4) | 66,751 (9.7) | 64,038 (16.5) |
|     Asian | 52,356 (8.3) | 33,106 (4.7) | 29,707 (7.6) |
|     Hispanic | 88,197 (13.9) | 77,846 (11.3) | 68,798 (17.7) |
| Education level (%) |  |  |  |
|     Less than 12th grade | 2,571 (0.4) | 2,766 (0.4) | 3,080 (0.8) |
|     High school diploma | 120,045 (19.0) | 147,990 (21.4) | 118,531 (30.5) |
|     Less than Bachelor's | 333,560 (52.8) | 373,284 (54.0) | 208,949 (53.7) |
|     Bachelor's degree plus | 175,161 (27.7) | 167,519 (24.2) | 58,697 (15.1) |
| Household income (%) |  |  |  |
|     <$40k | 95,517 (15.1) | 101,403 (14.7) | 103,685 (26.6) |
|     $40k - 49k | 35,275 (5.6) | 38,394 (5.6) | 33,165 (8.5) |
|     $50k - 59k | 36,644 (5.8) | 42,394 (6.1) | 35,179 (9.0) |
|     $60k - 74k | 57,411 (9.1) | 66,384 (9.6) | 46,627 (12.0) |
|     $75k - 99k | 94,409 (14.9) | 111,848 (16.2) | 62,877 (16.2) |
|     >$100k | 311,981 (49.4) | 331,136 (47.9) | 107,724 (27.7) |

