# OpenReview forum: "Automated Detection of Interpretable Causal Inference Opportunities: Regression Discontinuity Subgroup Discovery"
_ICML.cc/2023/Workshop/IMLH — IMLH 2023 Poster_

### Official Review · Reviewer_ebDD · 2023-06-06

**Rating:** 7
**Confidence:** 3

**Review:**

The paper suggest an algorithm for subgroup discovery in Regression discontinuity tasks. The paper is both well written and interesting. To the best of my knowledge the proposed algorithm is novel enough and without having checked the math, it appears good. Overall i think it would be a good addition to the set of papers of this workshop

---

### Official Review · Reviewer_2gPG · 2023-06-18
**Good paper for regression discontinuity discovery by considering subgroup effects**

**Rating:** 7
**Confidence:** 3

**Review:**

This paper identifies the heterogeneity in treatment assignment for subgroups and proposes a novel data-driven method, Regression Discontinuity Subgroup Discovery (RDSGD), to learn RD subgroups with different treatment assignment thresholds, which exhibits improved interpretability and statistical power upon previous methods. Experiment results show that taking subgroups into consideration when deciding RD is very useful. I recommend acceptance for this paper.

---

### Official Review · Reviewer_WW6J · 2023-06-18
**A clinically relevant approach of causal inference**

**Rating:** 7
**Confidence:** 2

**Review:**

This work introduces a method to automatically detect casual inference opportunities. The authors introduce novel approaches and illustrate them in multiple clinical contexts. Overall, the paper has multiple strengths - the details of the method, the theoretical background and clinical experiments.

---

### Meta-Review · Area_Chair_YhtU · 2023-06-18

**Recommendation:** Accept (Poster)
**Confidence:** 4

**Metareview:**

This paper identifies the heterogeneity in treatment assignment for subgroups and proposes a novel data-driven method, Regression Discontinuity Subgroup Discovery (RDSGD), to learn RD subgroups with different treatment assignment thresholds.

All reviewers found the paper sound and interesting, presenting a timely contribution to interpretable ML methods in healthcare.

---

### Decision · Program_Chairs · 2023-06-20

Accept (Poster)